# Thermal cues drive plasticity of desiccation resistance in montane salamanders with implications for climate change

Eric A. Riddell [1], Emma Y. Roback [1,2], Christina E. Wells [1], Kelly R. Zamudio [3] & Michael W. Sears[1]

Organisms rely upon external cues to avoid detrimental conditions during environmental change. Rapid water loss, or desiccation, is a universal threat for terrestrial plants and animals, especially under climate change, but the cues that facilitate plastic responses to avoid desiccation are unclear. We integrate acclimation experiments with gene expression analyses to identify the cues that regulate resistance to water loss at the physiological and regulatory level in a montane salamander (*Plethodon metcalfi*). Here we show that temperature is an important cue for developing a desiccation-resistant phenotype and might act as a reliable cue for organisms across the globe. Gene expression analyses consistently identify regulation of stem cell differentiation and embryonic development of vasculature. The temperature-sensitive blood vessel development suggests that salamanders regulate water loss through the regression and regeneration of capillary beds in the skin, indicating that tissue regeneration may be used for physiological purposes beyond replacing lost limbs.

[1] Department of Biological Sciences, Clemson University, 132 Long Hall, Clemson, SC 29631, USA. [2] Biology Department, Grinnell College, 1116 Eighth Ave, Grinnell, IA 50112, USA. [3] Department of Ecology and Evolutionary Biology, Cornell University, E145 Corson Hall, Ithaca, NY 14853, USA. Correspondence and requests for materials should be addressed to E.A.R. (email: riddell@berkeley.edu)

Organisms rely upon their ability to sense and integrate cues from their external environment to maintain fitness during environmental change[1–4]. At short time scales, these cues facilitate phenotypic changes by stimulating gene expression to regulate fitness-related traits[5]. In turn, the response helps to anticipate fluctuating resources[6] and avoid conditions that threaten survival[7,8]. These environmentally mediated responses are a widespread strategy among organisms termed phenotypic plasticity[9]. Plasticity allows individuals to adjust behavioral, morphological, or physiological traits within their lifetime in response to environmental conditions[1]. By operating within individuals, plasticity acts on faster time scales than evolutionary responses and reduces reliance on dispersal, which can be energetically-demanding or infeasible[10]. Due to its rapid and reversible nature, physiological plasticity has the potential to limit the loss of global biodiversity during severe environmental change[11–13], such as increased desiccation risk from climate change[14,15]. Yet, we lack knowledge on the cues and underlying genetic pathways responsible for avoiding the physiological and ecological consequences of a warming and drying environment.

Thermal cues are commonly used by plants and animals due to its association with fitness-related biotic and abiotic variables[16], such as prey availability[6] and photosynthetic performance[17]. Temperature has the potential to act as a reliable cue for desiccation risk because warming triggers greater evaporation rates by increasing the vapor pressure deficit (VPD), the primary physical factor driving rates of evaporative water loss[18]. VPDs increase with warming because the amount of vapor required to saturate the air (termed the saturation vapor pressure) increases exponentially with temperature[19], and therefore, might represent a cue for greater desiccation risk. High VPDs can impose strong selective pressure by causing large-scale mortality in plants[15,20] and animals[14,21] under climate change. Plasticity in water loss rates, in turn, provides an adaptive strategy for plants[22] and animals[23,24] to reduce desiccation risk. Using thermal cues as opposed to hydric also allows organisms to avoid desiccation, rather than responding to the physiological consequences (i.e., dehydration) after they have begun[25]. To date, studies have yet to evaluate the potential for organisms to use temperature as a cue to predict desiccation risk and how these responses are regulated at the genetic level. Identifying the genetic mechanisms underlying plasticity can reveal targets of selection during environmental change[26–29], improving predictions on the ecological and evolutionary outcome of climate change.

Here, we identify the cues and functional gene networks underlying plasticity of skin resistance to water loss ($r_i$) in a fully terrestrial, montane salamander (*Plethodon metcalfi*). The cues and mechanisms are particularly relevant for salamanders because resistance to desiccation determines the spatial distribution of their fundamental niche[30,31] and regions with the highest extinction risk under climate change[13]. In the present study, we identify the potential for temperature to act as a cue for high and more variable VPDs at local and global scales. We explicitly assess temperature, VPD, and their interaction as cues for plasticity at the physiological and regulatory level in a month-long acclimation experiment with a randomized, full factorial experimental design. We use a high throughput RNA-seq approach to sequence total RNA from experimental salamanders, build a de novo transcriptome, and link gene expression in the skin to phenotypic changes using multiple independent analyses on gene expression. Our analysis includes targeted gene ontology terms based upon the mechanisms known to influence plasticity in amphibian water loss physiology, specifically involving lipid barrier formation[32] and regulation of the skin's vasculature[33]. We find that temperature is an important cue for regulating water loss rates in salamanders and might act as a similar cue for plants and animals

across the globe. We also discovered that *P. metcalfi* appears to regulate $r_i$ using temperature-dependent regression and regeneration of bloody vessels in their skin.

## Results

**Desiccation risk rises exponentially with temperature.** Salamanders experienced an exponential rise in desiccation risk as temperatures warmed in the field (Fig. 1). We measured night-time temperatures and VPDs on the forest floor during the months of salamander activity (April to October) in 2013, 2015, and 2016 using iButtons (see Methods). Physical expectations suggest that warming temperatures increase VPDs and variability in the VPD due to the exponential rise in vapor required for saturation ($e_s$), that is 100% relative humidity (Fig. 1a). VPDs were higher in the late spring, but the air became more saturated during the summer and into early fall (Fig. 1b). Non-linear regression analyses that controlled for site-specific and annual variation revealed that the VPD increased exponentially with air temperature (Fig. 1c, Supplementary Table 1). Warming temperatures also coincided with greater variability in VPDs, especially after 20 °C (Fig. 1d, Supplementary Table 1). The rapid increase of the evaporative demand of the air on warm nights indicates that desiccation risk increases dramatically with warming.

**Thermal cues across the globe.** We conducted an analysis on the *TerraClimate* database to evaluate the potential for temperature to act as a reliable cue for desiccation risk across the globe[34]. We analyzed the associations between temperature, VPD, and the standard deviation of VPD over the last 30 years to determine hotspots of thermal cue reliability. Our global analysis revealed that temperature is associated with VPD and variation in VPD across much of the globe's terrestrial ecosystems (Fig. 2). Most regions between 50° and −50° latitude exhibited a positive association with temperature and VPD (Fig. 2a). This positive relationship was especially high in the Americas, Africa, and Australia; however, temperature may be a poor predictor of desiccation risk in the tropics due to high error in the relationship (Fig. 2b). Warm temperatures were also associated with greater variability in VPD, with two exceptions (Fig. 2c). In the Saharan Desert and Tibetan Plateau, warmer temperatures were associated with less variability in VPD. Sub-Saharan Africa exhibited the greatest uncertainty between temperature and the variability in VPD (Fig. 2d). These global relationships were consistent with empirical measurements in the Appalachian Mountains and predictions based upon theory.

**Individual variation in plasticity in $r_i$.** We measured plasticity in water loss physiology by evaluating an individual's change in $r_i$ over a 4-week experiment in response to ecologically relevant temperatures and VPDs. The experiment consisted of nightly exposures to a combination of cycling temperatures and humidity in a full factorial design across four treatments (Supplementary Fig. 1). The experiment was specifically designed to evaluate whether salamanders use temperature, VPD, or their interaction as cues to regulate $r_i$. The capacity to adjust $r_i$ was influenced by mass, temperature, and an individual's initial value of $r_i$ in the experiment. Salamanders that began the experiment with low $r_i$ (roughly 4 s cm$^{-1}$) exhibited the greatest increase in $r_i$ by the end of the experiment (Fig. 3; as determined by linear regression analysis [LRA], $\beta = -0.694 \pm 0.0774$, $p = < 0.001$, $\omega^2 = 0.378$). These individuals increased $r_i$ by as much as 68% during the experiment until they reached a maximum resistance (roughly 7 s cm$^{-1}$). Salamanders with a high initial $r_i$ exhibited a low capacity to adjust $r_i$, suggesting they had reached a physiological limit. We

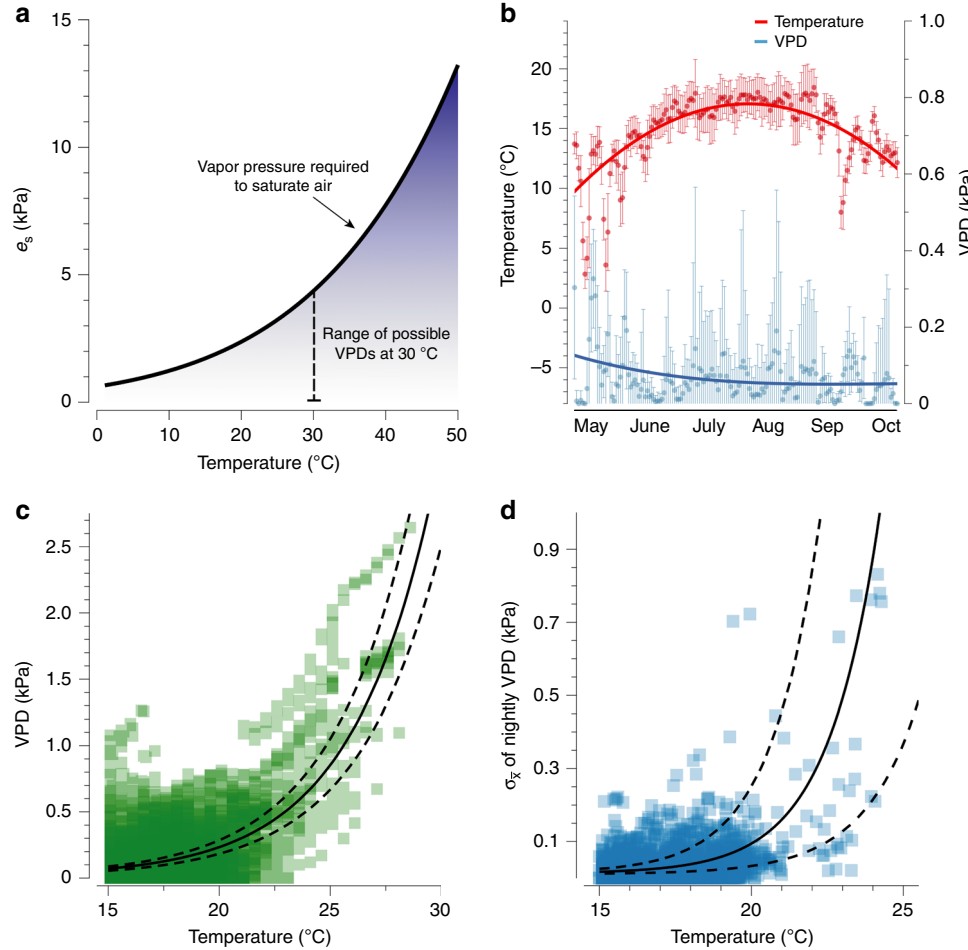

**Fig. 1** The risk of desiccation increases exponentially with warming. **a** Physical relationship between saturation vapor pressure ($e_s$) and air temperature indicates that VPDs are likely to become drier and more variable as climates warm. Blue shading indicates more water required to saturate the air, whereas white indicates less water required for saturation. **b** Seasonal variation in average nightly temperature and VPD (with standard deviations) from dataloggers during peak times of the year for surface activity in salamanders. Temperature (right axis) is displayed in red and VPD (left axis) is displayed in blue. **c**, **d** Non-linear regression analyses with 95% confidence intervals indicated that VPDs (**c**) and the nightly variation in VPD (**d**) increase exponentially with warming

also discovered an interaction between mass and temperature in which large individuals increased $r_i$ under warm temperatures more than large individuals in cool temperatures (Fig. 2; $\beta = 0.477 \pm 0.141$, $p < 0.001$, $\omega^2 = 0.021$). Inclusion of the mass by temperature interaction improved model fit by 5%. We removed VPD from the analysis because it and its interactions failed to explain any variation in $r_i$ ($p > 0.05$ and $\omega^2 < 0.01$).

**The de novo transcriptome for *P. metcalfi*.** We extracted total mRNA from salamander skin tissue before and after our experiments, sequenced transcripts on an Illumina platform, constructed a de novo transcriptome using Trinity (v. 2.4.0), and compared gene expression among treatments. Our sequencing results produced a high-quality de novo transcriptome for *P. metcalfi*. Trimmomatic and ConDeTri reduced raw sequences by 12.8%, resulting in a final number of 826 million reads with an average of 17.2 million reads per sample after filtering and trimming. Average base pair quality was consistently greater than 30 along the entire sequence for each sample based upon the FastQC files. All adapters and overrepresented sequences were successfully removed. In total, our assembled de novo transcriptome consisted of over 72,700 genes and gene isoforms. Transdecoder identified 17,300 of these genes as likely coding

regions. After further filtering for gene counts, we proceeded with 13,763 genes for differential gene expression analysis.

**Differential gene expression reveals temperature as a cue.** Our differential gene expression (DGE) anaylsis indicated that salamanders altered gene expression in the skin in response to temperature, but not VPD. Using the most liberal significance threshold ($\alpha = 0.1$), we found that 203 genes were upregulated and 308 genes were downregulated in the warm treatment relative to the cool. Only five genes were downregulated in response to high VPDs, and no genes were upregulated. For genes downregulated in response to warm temperatures, we found significant enrichment for 37 biological processes, 6 cell components, and 17 molecular functions (Supplementary Table 2). In general, downregulated genes were associated with stem cell differentiation (GO:0048863, as determined by GO enrichment analysis [GOEA], $p = 1.18 \times 10^{-4}$), endothelial cell differentiation (GO:1901796, GOEA, $p = 1.66 \times 10^{-2}$), and processes related to spleen, thymus, and liver development (GO: 0048536, GOEA, $p = 7.74 \times 10^{-5}$; GO: 0048538, $p = 5.01 \times 10^{-4}$; GO:0001889, $p = 1.03 \times 10^{-2}$). For upregulated genes in warm treatments, we found enrichment for 30 biological processes, 13 cell components, and 19 molecular functions (Supplementary Table 3). Common trends among these terms included regulation of NF-kappa $\beta$

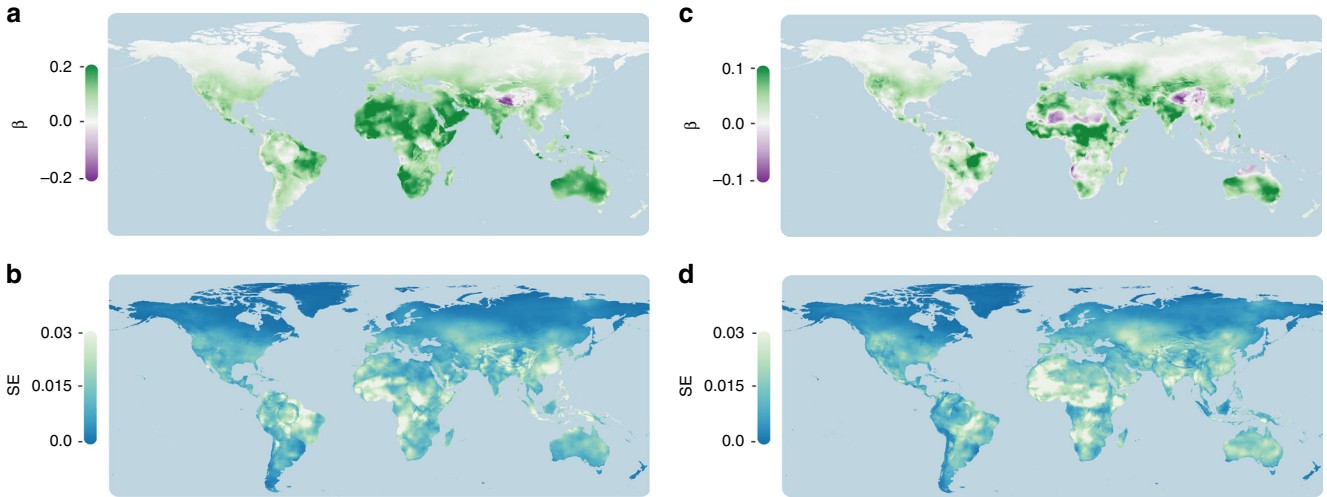

**Fig. 2** Global hotspots of temperature as a reliable cue for desiccation risk. **a** Temperature was generally positively correlated with VPD (as indicated by the slope (ß)). **b** The correlations between temperature and VPD is generally strongest between 50° and −50° latitude (determined using standard error (SE) of the slope), with regions of high correlation in South America, Africa, the Middle East, and Australia. **c** Temperature was generally positively associated with the variation in VPD. **d** The standard error of the correlations between temperature and variation in VPD is generally low outside of the tropics but high for areas in Africa and parts of the Amazon and south-central Asia. Figures were generated using custom script in Python (v. 3.6) using data from the *TerraClimate* database (http://www.climatologylab.org/terraclimate.html)[34]

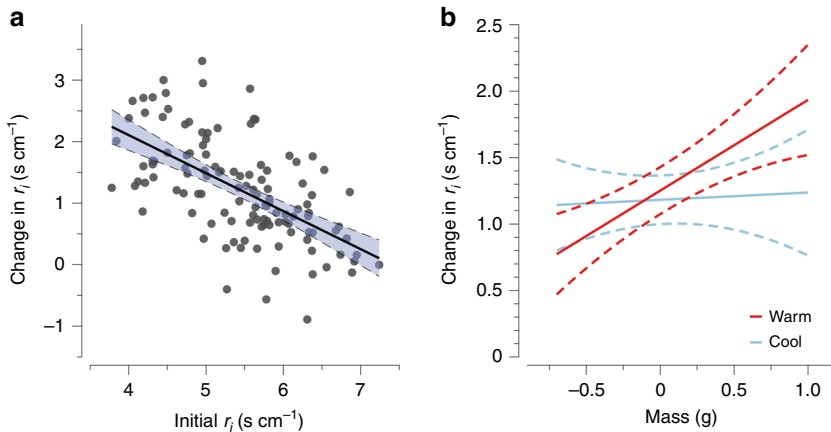

**Fig. 3** Plasticity of $r_i$ in response to initial $r_i$, mass, and temperature. **a** The spectrum of plastic responses in which individuals with low $r_i$ at the beginning of the experiment (Initial $r_i$) exhibited the greatest increase in $r_i$ by the end of the experiment and individuals with high $r_i$ did not exhibit plasticity. **b** Large individuals in the warm treatment (in red) increased $r_i$ to the greatest extent relative to individuals in the cool treatment (in blue). Mean responses with 95% confidence intervals are plotted

transcription factor (GO:0051992, GOEA, $p = 1.29 \times 10^{-3}$), protein folding (GO:0006457, GOEA, $p = 3.66 \times 10^{-3}$), response to hydrogen peroxides (GO:0042542, GOEA, $p = 1.01 \times 10^{-2}$), and responses to heat (GO:1900034, GOEA, $p = 1.23 \times 10^{-2}$). In general, we found that downregulated genes were related to stem cell differentiation and developmental processes, and upregulated genes were related to stress responses associated with warm temperatures.

Lowering the significance threshold of our analyses resulted in similar patterns. At the significance threshold of 0.001, we found four genes that were strongly downregulated in the warm treatments related to Jumonji isoforms (Supplementary Table 4), a suite of genes involved in regulation of stem cell differentiation[35] and blood vessel growth[36]. We also identified four genes associated with heat shock proteins and responses to stress (Supplementary Table 4). No matter the stringency of the significance threshold, our DGE and GO term enrichment analyses suggested that temperature represents a cue for the

downregulation of stem cell differentiation and developmental processes.

For genes downregulated in response to dry VPDs, three of the five genes were uncharacterized. The other two genes are involved in heat shock responses (GO:000948) or regulation of responses to pheromones (GO:0019236). These genes are the DNAj homolog subfamily A member 4 (*DNAJA4*, $p_{adj} = 1.42 \times 10^{-2}$) and the vomeronasal type-2 receptor 26 (*Vmn2r26*, $p_{adj} = 1.05 \times 10^{-3}$), respectively.

**Network analysis indicates role of blood vessels and lipids**. We used a weighted gene co-expression network analysis (WGCNA) to identify suites of gene networks that were associated with plasticity of $r_i$. These gene networks can identify important functional groups of genes and candidate genes underlying phenotypic responses[28,37,38]. We identified gene networks relevant to plasticity (termed *skin resistance modules*) based upon a module's

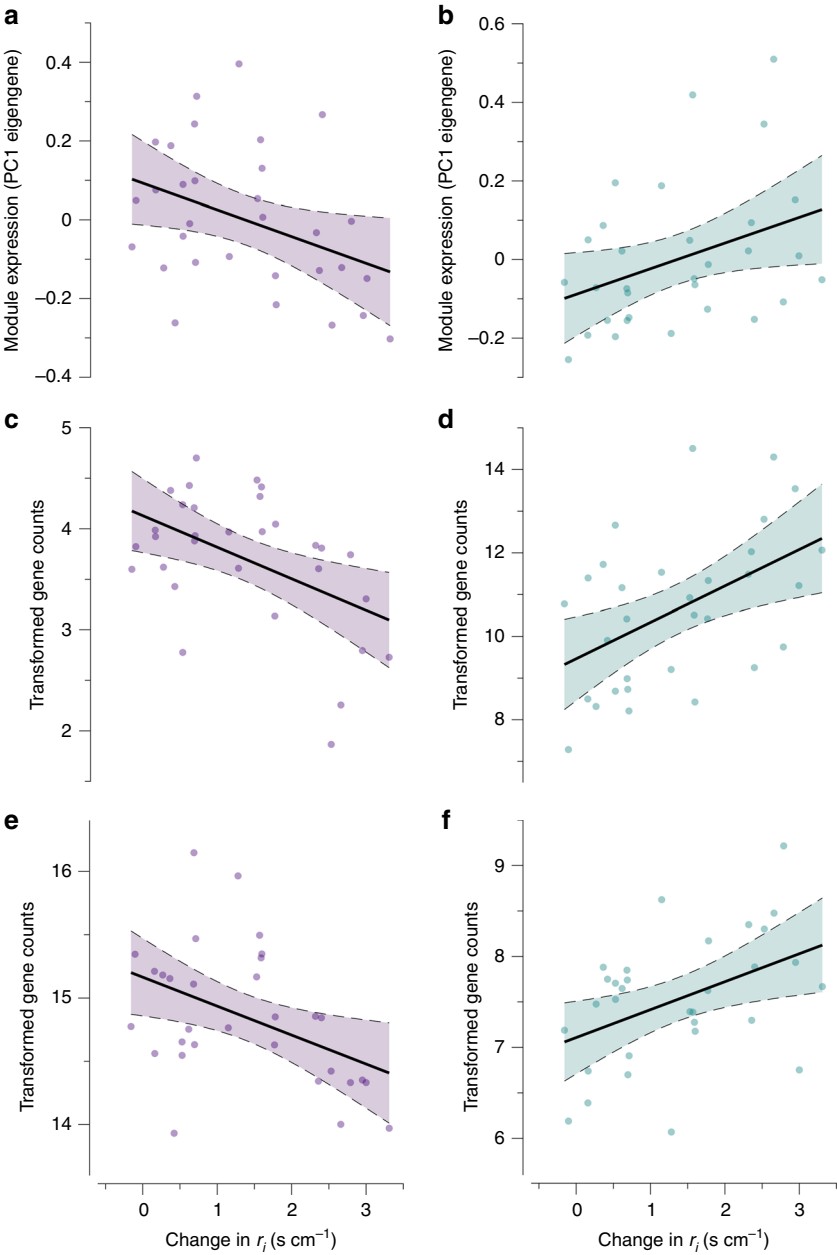

**Fig. 4** Skin resistance modules and candidate genes associated with plasticity in $r_i$. **a** The module negatively associated with $r_i$. **b** The module positively associated with $r_i$. These modules provided insight into the mechanisms underlying acclimation of $r_i$ by guiding subsequent analyses on gene ontology enrichment and hub gene identification. The figure also identifies four genes (*CHRM3* (**c**), *TRIM63* (**d**), *ALOXE3* (**e**), and *PDE4B* (**f**)) within the skin resistance modules that correlated to plasticity in $r_i$ and were associated with GO terms related to water loss regulation. Temperature and humidity were not associated with the expression of these modules. Regressions are shown with 95% confidence intervals

association with the change in $r_i$ over the course of the experiment. Our analysis identified 26 gene networks or modules. One of these modules was negatively correlated with $r_i$ (Fig. 4a), and the other was positively correlated with $r_i$ (Fig. 4b). The positively correlated module consisted of 203 genes, and the negatively correlated module consisted of 75 genes (Supplementary Table 5). Temperature and humidity did not influence gene expression in either module, but expression of both modules was associated with the change in $r_i$ (Supplementary Tables 6 and 7). These analyses revealed that module expression explained 11% and 10% of the change in $r_i$ for the negatively and positively associated modules, respectively.

The GO term enrichment analysis of genes within skin resistance modules identified several developmental processes,

similar to the DGE analysis. We found enrichment for 18 biological processes, 5 cell components, and 7 molecular functions (Supplementary Table 8). Our analysis revealed enrichment of genes associated with angiogenesis (GO:0016525, GOEA, $p = 3.37 \times 10^{-3}$), long-chain fatty acid metabolic process (GO:0001676, GOEA, $p = 2.59 \times 10^{-2}$), and branching involved in blood vessel morphogenesis (GO:0001569, GOEA, $p = 3.31 \times 10^{-2}$). We identified two genes, semaphorin (*SEMA*) and semaphorin-3E (*SEMA3E*), that were associated with enriched GO terms for angiogenesis and also within the skin resistance modules. These two genes were not correlated to the change in $r_i$ (as determined by weighted gene co-expression analysis [WGCNA], $p = 0.396$ and $p = 0.442$, respectively); however, they were identified as important hub genes within the skin resistance

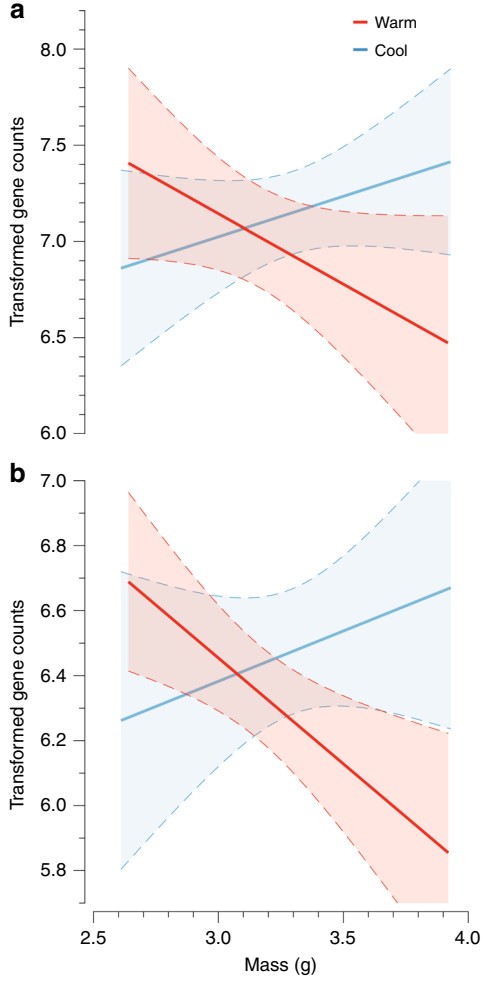

**Fig. 5** Blood vessel formation genes within skin resistance modules related to variation in mass and temperature. The figure illustrates two genes within the skin resistance modules that exhibit an interaction between mass and temperature. The temperature treatments are also identified in red (warm) and blue (cool) clearly illustrating an effect of temperature treatment. The figure illustrates the interaction for stromal interaction molecule 1 (*STIM1*) (**a**) and rap guanine nucleotide exchange factor 1 (*RAPGEF1*) (**b**). These genes provided insight into the mechanisms underlying temperature as a cue for $r_i$ as they both have the same role in blood vessel development. Regressions are shown with 95% confidence intervals

modules ($MM = -0.603$, $p < 0.001$ and $MM = -0.693$, $p < 0.001$, respectively).

Our co-expression network analysis also identified hub genes associated with muscle contraction and skin barrier development that correlated to plasticity in $r_i$. The network analysis revealed 13 hub genes that were positively correlated with the change in $r_i$ (Supplementary Table 9). Two of these hub genes are associated with muscle contraction (GO:0006936, GO:0010614, GO:1901898, GO:0086006), and these genes represented the first and third highest correlation to plasticity in $r_i$. These genes were the E3 ubiquitin-ligase (*TRIM63*, Fig. 4d, WGCNA, $r = 0.487$, $p = 4.65 \times 10^{-3}$) and the cAMP-specific 3,5-cyclic phosphodiesterase (*PDE4B*, Fig. 4f, WGCNA, $r = 0.443$, $p = 1.24 \times 10^{-2}$), both of which had high and significant module connectivity (Supplementary Table 9). We also found 19 hub genes that were negatively correlated with the change in $r_i$. Of these, the highest ranking hub gene, muscarinic acetylcholine receptor (*CHRM3*), was negatively correlated with the change in $r_i$ (Fig. 4c, $r =$

$-0.482$, WGCNA, $p = 5.96 \times 10^{-3}$) and exhibited high connectivity ($MM = 0.628$, WGCNA, $p < 0.001$). We also identified the hydroperoxide isomerase (*ALOXE3*) as having the fourth highest negative relationship with the change in $r_i$ (Fig. 4e, $r = -0.434$, WGCNA, $p = 1.47 \times 10^{-2}$) and high connectivity ($MM = 0.733$, WGCNA, $p < 0.001$). These two negatively correlated hub genes are associated with the regulation of smooth muscle contraction (GO:0045987 and GO:003056) and establishment of a skin barrier for limiting water loss (GO:0061436), respectively.

**Thermal cues regulate blood vessel growth**. Down-regulation of genes related to blood vessel branching and development under warm temperatures coincided with greater resistance to water loss (Fig. 5). We identified candidate genes underlying the response to temperature by isolating genes within the skin resistance modules that exhibited the same interaction between mass and temperature as $r_i$. For this *post hoc* analysis, termed the *interaction analysis*, we identified six genes from the skin resistance modules that exhibited a significant interaction with mass and temperature. Two of these genes, stromal interaction molecule 1 (*STIM1*) (Fig. 5, as determined by analysis of covariance [ANCOVA], $p = 0.0277$) and rap guanine nucleotide exchange factor 1 (*RAPGEF1*) (Fig. 5, ANCOVA, $p = 0.00247$), were associated with GO terms relating to plasticity of $r_i$. *STIM1* is associated with positive regulation of angiogenesis (GO:0045766)[39,40], and *RAPGEF1* is associated with positive regulation of vasculogenesis (GO:2001214) and blood vessel development (GO:0001568)[41]. These results suggest that blood vessel development underlies plasticity in $r_i$.

**Gene set enrichment analysis**. We identified up- and down-regulation of predefined physiological pathways in response to temperature and humidity using the Gene Set Enrichment Analysis (GSEA v. 2.1.0). In response to temperature, we found a suite of upregulated pathways involved in muscle contraction and downregulated pathways involved in thymus and liver development (Supplementary Table 10). The genes associated with these GO terms are known to be specifically involved in regulating blood vessel growth and stem cell differentiation (see below). For humidity, we found two upregulated pathways also involved in muscle contraction (though fewer compared to temperature) and downregulation of pathways involved in signal transduction and developmental processes (Supplementary Table 11).

**High degree of overlap between transcriptional approaches**. We determined the degree of overlap between independent transcriptional analyses to evaluate the robustness of our approach. The GSEA and WGCNA analyses shared eight genes (Supplementary Table 12) and the GSEA and DGE analyses also shared 17 genes (Supplementary Table 13). For genes shared between GSEA and WGCNA analyses, two genes were significantly related to plasticity in $r_i$. These two genes, JAK2 tyrosine kinase (*JAK2*) and phosphodiesterase-4B (*PDE4B*), are involved in proliferation of stem cells[42] and muscle contraction, respectively. *PDE4B* was also identified in the WGCNA analysis as an important hub gene related to plasticity in $r_i$ (Fig. 4f). For the GSEA and DGE overlap, we found several genes involved in stem cell differentiation related to the Jumanji isoforms that were down-regulated in response to warm temperatures (Supplementary Table 13). These genes are also known to be involved in blood vessel growth[36]. We also found several genes related to muscle contraction (*UTRN*), cell differentiation (*UNC45B* and *BCL6*), and angiogenesis (*Cav1*). We found four genes that overlapped between the WGCNA and DGE analysis

(Supplementary Table 14), which involved genes that regulate inhibition of blood flow (hemostasis), lipid synthesis, and protein transport. Interestingly, the gene related to hemostasis (APLP2) was also correlated with plasticity in $r_i$ (Supplementary Table 14). Assessing overlap revealed robust, common trends between the independent gene expression analyses that supported the role of blood vessel development and lipid barriers in regulating $r_i$.

## Discussion

Our analysis revealed a novel perspective on the physiological significance of tissue regeneration. Gene expression analyses suggested that the high resistance to water loss phenotype (and thus low plasticity phenotype) were associated with the constitutive expression of ALOXE3, a gene involved in the enzymatic pathway for regulating transepidermal water loss. At the other end of the spectrum, the high plasticity phenotype was associated with genes that regulate vasoconstriction and blood vessel development. Without the continued expression of ALOXE3, salamanders appeared to adjust $r_i$ by limiting blood flow to the skin's vasculature, an effective strategy to reduce water loss[33]. Not only did salamanders regulate pathways involved in vasoconstriction, hemostasis, and angiogenesis, but also pathways involved in vasculogenesis, the de novo formation of capillaries from hematopoietic stem cells typically confined to embryos[43]. Salamanders, however, are known for their ability to regenerate many types of tissues by continuing to express embryonic developmental pathways[44,45]. Thus, salamanders may adjust water loss across their skin using their unique ability to regenerate tissues by regulating dermal capillary growth in response to environmental change.

From the broad functional perspective, independent enrichment analyses revealed consistent involvement of genes, gene networks, and known gene pathways related to developmental processes and stem cell differentiation. Each analysis identified the regulation of muscle contraction, blood vessel growth, and stem cell proliferation, suggesting that regeneration of capillary beds is involved in the temperature-sensitive plasticity of resistance to desiccation. We hypothesize that repeated exposure to warm temperatures elicits vascular regression in the capillary beds of the skin. The process of vascular regression begins with sustained vasoconstriction, eventually leading to hemostasis and the programmed cell death of capillary beds over the course of several weeks[46]. Once the threat of desiccation has passed, salamanders might then regenerate dermal capillary beds using vasculogenesis and angiogenesis to return blood flow. Given the potential for plasticity in $r_i$ to reduce extinction risk from climate change[13], our study draws connections between the regenerative capacity of salamanders and the ecological consequences of increasing $r_i$ through physiological plasticity.

The expression of genes related to tissue regeneration may also have alternative explanations, but we consider these less likely. Amphibians shed more frequently in response to warming[47], and the growth of new skin would likely influence blood vessel growth. However, increased rates of shedding in amphibians are associated with higher rates of water loss[48], but in our study, we found the reduction in water loss rates (or increase in $r_i$) was associated with expression of genes related to angiogenesis and stem cell differentiation. The conflicting patterns suggest that the thermal sensitivity of shedding is unlikely to underlie the observed gene expression patterns. Being lungless, these salamanders might also regulate blood vessel formation to adjust oxygen and carbon dioxide exchange across their skin. Recent evidence, however, suggests that plasticity in $r_i$ and gas exchange are tightly coupled[31]. Vascular regression in the skin would thus simultaneously reduce water loss rates and gas exchange, a pattern recently observed in this species[31]. The subsequent reduction in metabolic rate likely represents a consequence associated with reducing water loss rates by adjusting the capillary densities in the skin. The exact role of vasculature regeneration requires studies that explicitly link physiological plasticity to more direct cellular and molecular evidence of new capillary growth in the skin.

We observed a high degree of individual variation in the capacity to regulate resistance to water loss in our experiments. Individuals that exhibited a high initial $r_i$ (roughly 7 s cm$^{-1}$) maintained a high $r_i$ throughout the entire experiment, exhibiting limited plasticity. The maintenance of high $r_i$ was unexpected given that salamanders were maintained in saturated conditions for a month prior to the experiment, and consequently, this upper boundary may be indicative of a physiological limit. At the other end of the spectrum, some individuals began the experiment with a relatively low $r_i$ (4 s cm$^{-1}$) and increased $r_i$ over the course of the experiment by as much as 68% to the maximum value. Thus, the population consists of individuals that fall across a spectrum of plasticity, which raises intriguing questions about the benefits of plasticity or lack thereof. Low $r_i$ would predispose individuals to desiccation. However, a low resistance might also increase metabolic scope by promoting greater oxygen uptake[31]. Thus, individuals might benefit from low resistance by supporting oxygen-demanding activities, such as foraging or defending territories. At the other end of the spectrum, limiting plasticity might reduce costs of maintaining sensory pathways[49,50] or maximal resistance in the face of unpredictable VPDs. The costs and benefits of water loss plasticity might be better understood by exploring these interactions.

Lipids are the fundamental unit that structures the resistance of the integument or cuticle to water loss[32]. The role of lipid regulation, though unexplored in salamanders, is consistent with mechanisms underlying water loss plasticity in other vertebrates. The composition and maintenance of lipid barriers plays a role in reducing water loss in amphibians[32], birds[51–53], and mammals[54]. Our study revealed a negative association with plasticity in $r_i$ and expression of the ALOXE3, a gene associated with production of ceramides, a type of essential fatty acid in epidermal tissue[55]. Loss of function mutations in this family of genes results in pathological disorders associated with high transepidermal water loss[55], and many mammals and birds rely on ceramides or related fatty acids to limit transepidermal water loss[52]. With lower production of ceramides, salamanders might then rely on vascular regression to regulate water loss. Thus, salamanders can achieve their relatively water-tight phenotype through two different pathways: lipid barrier formation or regulation of blood flow to the skin. Our study highlights the potential for complex interactions among underlying mechanisms in a system that has historically been characterized as one-dimensional[56].

Seminal studies on amphibian water loss physiology have suggested that many amphibians, and salamanders in particular, do not have a physiological resistance to water loss[56]. The assumption that most amphibians exhibit water loss rates similar to free water has been perpetuated for decades, often citing the seminal study on a single species of salamander[57,58]. This reification on amphibian physiology resulted in many studies overlooking variation in water loss physiology or focusing on rare forms of desiccation resistance, such as cocoon formation and waxy secretions[32,59]. Our results strongly conflict with the conclusion that salamanders do not have a resistance to water loss due to the observed plasticity in $r_i$. Moreover, these changes in resistance, though small relative to other animals, can have major implications for the fundamental niche of terrestrial salamanders[30]. Our study differs from previous evaluations of water loss by carefully measuring changes in vapor pressure contributed

by the animal over several weeks under ecologically relevant conditions using an advanced flow-through system. Thus, we strongly advocate for studies that investigate amphibian water loss physiology within and between species to promote the discovery of novel physiological variation and their underlying mechanisms.

We revealed temperature, but not humidity, as a cue for the regulation of water loss physiology at the regulatory and physiological level. Due to the unpredictable variation in VPD and the physical relationship between VPD and temperature (Fig. 1), our results provide a clear understanding for why selection may favor temperature as a cue for regulating water loss physiology. At global scales, temperature and VPD are highly correlated (Fig. 2), suggesting that many organisms across the globe likely use temperature as a cue to reduce water loss rates in anticipation of drier conditions, especially in temperate climates. Temperate ectotherms[60] and plants[61] support this hypothesis. In tropical climates, however, high error in the relationship between temperature and desiccation risk suggests that few organisms would use temperature as a cue to predict desiccation risk, and in support of this, some tropical ectotherms exhibit a limited capacity to adjust water loss rates[62]. In regions like the Tibetan plateau, organisms might use warming to anticipate more humid conditions due to the negative correlation between temperature and VPD. These global correlations can inform hypotheses for experiments that explore thermal cues across the globe and taxa.

Plants and animals also have different plastic responses to warming depending on the time scale of the exposure. In response to warming over short time scales, plants and animals often increase water loss rates to offload excess heat through evaporative cooling[63–65]. When exposed to warm temperatures during development or over longer periods of time, as in our study, organisms frequently conserve water by reducing water loss rates (or increasing resistance) to lower desiccation risk[61]. Thus, organisms experience a trade-off between cooling and conserving water that determines variation in water loss rates over different time scales[66,67]. Integrating these temporal considerations with the spatial variation in thermal cues will improve predictions on plastic responses to climate warming across a wide range of taxa.

Predicting ecological responses to novel environmental conditions depends upon understanding plasticity of environmentally sensitive traits. However, the role of plasticity in novel environments is contentious. In some cases, plastic responses can facilitate rapid phenotypic evolution[6,29,68], whereas in other cases, plasticity limits the ability to respond to adapt to novel environments[69]. Despite these uncertainties, our study provides important insight into the use of temperature as a cue for plasticity, the ecological significance of tissue regeneration, and the complex mechanisms underlying water loss plasticity. Evaluating these mechanisms as targets of selection across space, time, and taxa has the potential to inform their evolutionary potential in a rapidly warming climate.

## Methods

**Salamander collection**. We collected 132 salamanders (*P. metcalfi*) from the Balsam Mountain Range in the Nantahala National Forest (35° 20′ N, 83° 4′ W) during May 2016. We collected no more than four salamanders per site, and sites were distributed in a randomized designed between high (1600 m), mid (1400 m), and low elevations (1200 m) to account for potential elevational variation in water loss rates[70]. Each collection point was generated using a random point generator on QGIS (v.2.1). We created a buffered region around the dirt access road to generate points at least 100 m away from the road to minimize possible road effects. We placed each salamander in an individual Ziploc® bag with moist leaf litter for transported back to Clemson University on the night of collection. Salamanders were housed in individual plastic containers (17 cm × 17 cm × 12 cm) with moist paper towels for rehydration throughout the month-long acclimation period. All salamanders were maintained in a Percival incubator (Percival, Inc.; Model #I-

36VL) under a cool, cycling thermal regime for 1 month to acclimate to laboratory conditions. The cycling thermal regime was designed to mimic conditions that salamanders experience in the early spring[31]. We rotated the position and shelf for each salamander every day so that salamanders experienced every combination of position (front, middle, back) and shelf location (1–4) during the course of the acclimation period. All individuals were fed crickets (*Acheta domesticus*) ad libitum. These protocols were also continued during the acclimation experiment, and in addition, we rotated treatments among incubators to reduce the chance of an incubator effect. We complied with all relevant ethical regulations for animal testing and research with approved protocols from the Institute for Animal Care and Use Committee at Clemson University (#2014-024). Field collections were approved by the North Carolina Wildlife Commission (#16-SC00746) and United States Fish and Wildlife Service (#MA90761B-0).

**Acclimation experiments**. The acclimation experiment was specifically designed to tease apart temperature and humidity as cues for physiological plasticity. After the 1-month acclimation period, salamanders were randomly assigned to a cycling thermal regime. The cool cycling thermal regime resembled the temperature cycle during the acclimation period, which fluctuated between 8 and 15 °C. The warm temperature cycle fluctuated between 15 and 22.5 °C. The cool temperature cycle was developed from conditions that salamanders experience in the early spring, and the warm cycle was developed from conditions that salamanders are likely to face under climate change assuming the Representative Concentration Pathway 8.5 —near a 5 °C increase in air temperature by the end of the century in the southern Appalachian Mountains[71]. Each temperature was also assigned a particular humidity treatment: dry or wet. The dry cycle maintained a VPD of 0.4 kPa, whereas the wet cycle maintained a VPD of 0.2 kPa. These VPDs correspond to humidities that salamanders experience in nature (Fig. 1). We maintained these VPDs during the thermal regime by cycling relative humidities throughout the day. Therefore, our experiment consisted of a full factorial design of four treatments: (1) warm, wet; (2) warm, dry; (3) cool, wet; (4) cool, dry. Prior to salamanders being exposed to any treatments, we measured baseline skin resistance to water loss rates using a flow-through system (see Flow-through system). After baseline measurements, we extracted total RNA from 12 randomly selected individuals, and the remaining 120 individuals were randomly assigned to their treatments with 30 individuals per treatment.

We exposed salamanders to their treatment by moving them to activity enclosures for a three-hour period between the hours of 2100 and 0600. Activity enclosures were the same size as their enclosures during the pre-experiment acclimation period, but they consisted of dry soil as a substrate. We dried the soil to make sure that variation in soil moisture did not influence the vapor content of the air. We also ensured that VPDs in the activity chambers reflected the desired vapor content of the treatment by placing three Hygrochron iButtons (Item# DS1923, Maxim Integrated, 160 Rio Robles, San Jose, CA 95134) on three randomly selected shelves every night of the exposure. The activity enclosures also had a hardwire mesh roof to allow air to freely circulate into the enclosure, exposing salamanders to the vapor content of the air. Salamanders were weighed prior to and after each activity simulation to ensure that they did not lose more than 10% of their baseline body mass, which we determined at the end of the 1-month acclimation period. In total, individuals were not exposed to their treatment 1.1% out of all possible exposures because their mass did not recover from the previous exposure. After five nights of exposure to their treatment, we measured $r_i$ using the flow-through system (see below). Immediately following the physiological measurement, salamanders were returned to their treatment and allowed to rest for one night. We continued this protocol (five nights of exposure, one night of physiological measurement, and one night of rest) for roughly 1 month. At the end of the experimental period, we sacrificed all salamanders to collect total RNA immediately following the final measurement in the flow-through system.

**Physiological measurements**. We measured $r_i$ using a flow-through system designed to carefully control temperature and humidity. Our flow-through system operated in push mode, starting with a sub-sampler pump (SS4; Sable Systems International, 3840 N. Commerce Street, North Las Vegas, NV 89032). The pump pulled air from within a temperature controlled Percival incubator that housed the chambers containing the salamanders. The air then flowed into a dewpoint generator (DG-4) to control the level of humidity (VPD = 0.5 kPa). The air then passed through a manifold to divide the airstream and control the flow rate (180 mL min⁻¹). The air then passed into the acrylic chambers (16 × 3.5 cm; volume ~153 mL) containing an individual salamander. Salamanders were suspended on top of a mesh platform in the chamber to ensure that salamanders could not behaviorally alter water loss rates by curling onto themselves. Their position was also intentionally made to mimic posture while walking on the forest floor. We ensured that each measurement was recorded from resting individuals by directly evaluating activity and ensuring vapor pressure recordings were stable—a pattern indicative of resting[72]. After the chamber, the air passed into a RH-300 to measure the vapor content of the air. Changes in voltage from each analyzer were continuously measured using Expedata (Sable Systems International). Equations for converting voltages into meaningful physiological values have been extensively reported in previous research[30,31]. Individuals were measured randomly across

treatments to ensure that a given round of measurements contained a random mixture of individuals across all treatments.

**Temperature as a cue for salamanders**. Identifying environmental cues underlying plasticity requires measurements of relevant abiotic conditions. We used iButton Hygrochrons (Item# DS1923) to simultaneously record temperature and humidity across collection sites from 2013 to 2016, excluding 2014. We recorded temperature and VPDs from May to October over this 3-year period ($n = 64,502$). We selected this monthly time period because it coincides with the activity season of _P. metcalfi_[73]. We evaluated the relationship between temperature and vapor pressure based upon the thermal preferences of montane salamanders, which occur between 15 and 25 °C[73]. Hygrochrons were deployed to randomly generated coordinates using similar methods as the collection sites (see above). Each Hygrochron was housed within a custom-built hardwire mesh cage (1 cm gauge) and secured within the cage using a plastic cable tie. The sensors were placed 1 cm from the ground to record temperatures and VPDs that salamanders experience while roaming the forest floor. Temperature and relative humidity were recorded every 20 min during the sampling period, and in most cases, temperature was recorded to the nearest tenth decimal to extend the sampling period by reducing the data file size.

**Temperature as a cue across the globe**. We assessed the relationship between temperature and VPD across terrestrial ecosystems to understand the potential for temperature to act as a cue for desiccation risk. We used estimates of monthly minimum and maximum temperatures and VPDs from the _TerraClimate_ dataset[34]. These values were downscaled from globally-distributed field station measurements and advanced general circulation models. These conditions are relevant to any plant or animal experiencing temperature and VPD roughly 1–2 m from ground level. Permission to adapt these maps for our purposes was granted under Creative Commons Attribution 4.0 International License (http://creativecommons.org/licenses/by/4.0/). We evaluated the association between temperature and humidity using downloadable raster datasets from the _TerraClimate_ database (http://www.climatologylab.org/terraclimate.html), which have been recently described[34].

We averaged monthly temperatures and VPDs to calculate annual temperature, annual VPD, and standard deviation of annual VPD for each location over a 30-year time period. We then developed custom code in Python (v. 3.6) to analyze the correlation between these environmental variables using linear regression analyses from the _statsmodels_ library. We sampled the spatial layers using the _gdal_ functions in _osgeo_ library to convert the spatial data into a large dataset (~60 gigabytes) for downstream analysis. Due to the large size of the dataset, we used the _chunksize_ parameter in the _pandas_ library to import portions of the global dataset for analysis. We then used the _multiprocessing_ library to evaluate correlations for each site (meaning, a single pixel of roughly 4 km) across 30 independent central processing units. These individual datasets were then compiled together and sorted into global maps using a custom function in Python. These maps were then visualized in QGIS (v. 2.1). All script is available at https://github.com/ecophysiology/global_cues.

**Extraction and sequencing of RNA**. We extracted total RNA from 12 individuals after the initial physiological measurements following the one-month acclimation period, but prior to any exposure to the treatments. These individuals are referred to as baseline individuals or samples. We then extracted total RNA from salamander skin tissue immediately following the final measurement of $r_i$. Individual salamanders were removed from the flow-through system and immediately immersed in liquid nitrogen for 15–20 min. After freezing, we used a flame-sterilized razor blade to shave off a thin layer of skin from the dorsal side of the salamander. We also extracted hearts from the frozen specimen using the flame-sterilized razor blade. The hearts were easily visible through the ventral side of the salamander because the ventral skin is translucent. Though heart samples were included in the de novo transcriptome, we did not evaluate gene expression differences because the analysis is beyond the scope of this study. The tissue samples were immediately immersed in TRIzol reagent (Item#: 15596026; Thermo Fisher Scientifics, 168 Third Avenue, Waltham, MA 02451) and blended using a motorized tissue homogenizer with sterilized pestle (Item#: UX-44468-25; Argos Technologies Inc., 625 E. Bunker Ct., Vernon Hills, IL 60061). Once fully homogenized, the tissue samples were stored at 80 °C until downstream preparation in accordance with standard protocols.

We extracted total RNA using standard protocols for TRIzol reagent. Prior to library preparation, we purified each sample to remove contaminants (e.g., phenol, ethanol, etc.) using the RNeasy Mini Kit for total RNA (ID: 74136; Qiagen®, 1001 Marshall St., Redwood City, CA 94063). In addition, we treated samples with DNase to limit the possibility of DNA contamination in tandem with the RNeasy Mini Kit. Prior to library preparation, we determined the concentration of total RNA using a Qubit Fluorometer (Thermo Fisher Scientific) to ensure each sample contained sufficient material for library preparation. We prepared each sample using the Illumina TruSeq® mRNA Stranded Kit (Product#: RS-122-2101, Illumina, Inc., 5200 Illumina Way, San Diego, CA 92122) by following standard protocols. Once again, we used the Qubit Fluorometer to quantify the amount of cDNA prior to sequencing.

We sequenced our samples using the Illumina HiSeq2500 platform at the Genomics Facility at Cornell University. We sequenced singe-end 100-bp reads for all libraries. We pooled 12 barcoded samples per lane to balance sequencing depth for each sample with the total number of samples ($N = 48$ total samples across four lanes). We randomly assigned an individual sample to each sequencing lane with respect to treatment to avoid lane biases. In preparation for de novo transcriptome assembly, we trimmed Illumina-specific sequences and adapters using Trimmomatic (v. 0.36). We evaluated error probabilities of each read with ConDeTri (v. 2.3) and low-quality bases or sections of multiple bases were removed using default parameters. After trimming, reads were assessed using FastQC (version 0.11.5) to confirm that extraneous reads had been removed and quality scores were greater than 30 along the entire sequence length.

We assembled our de novo transcriptome using 827 million reads from 48 tissue samples (36 skin and 12 heart samples) using Trinity (v. 2.4.0) and standard protocols for strand specific analysis and normalization. We used TransDecoder (version 3.0.1) to reduce the transcriptome to coding regions from all transcript sequences. TransDecoder identified likely coding sequences as those with a minimum open reading frame length of 50 amino acids. We then assigned gene ontology terms, enzyme codes, and KEGG pathways to genes by blasting against all databases using the Blast2GO Cloud Blast Function. We blasted against all known organisms to ensure the highest chance at annotating each gene. The number of blast hits was increased from the default parameter to 20, and the mapping and annotation steps were run using default protocols. RSEM (version 1.3.0) estimated gene and gene isoform expression levels by aligning individual samples back to the transcriptome using Bowtie (version 1.2.1.1). To prepare for differential gene expression analysis, low-expressed genes were filtered out, retaining only those for which approximately one-third of genes had 10 or more counts.

**Statistical analyses**. We analyzed the physiological data using _R_ (v.3.4.2) using linear regression analyses. We evaluated the change $r_i$ over the course of the experiment with mass and initial $r_i$ as covariates. Temperature and humidity were treated as factors. Change in $r_i$ was calculated by taking the difference of final $r_i$ and initial $r_i$ for each individual. We also evaluated interactions between covariates and factors to assess complex responses to temperature and humidity. Covariates were scaled and centered over zero, and we assessed collinearity between covariates using the _vif_ function from the _car_ package in _R_ to examine variance inflation factors (VIF). We excluded variables and combinations of variables that exceeded the VIF threshold of 3 (ref. [74]). We then conducted a Type-II analysis of covariance using the _car_ package to assess significance of covariates and factors. Finally, we report effect size using omega-squared ($\omega^2$) for each significant effect using the _sjstats_ package in _R_.

We analyzed the environmental data using non-linear, mixed effects models from the _nlmer()_ function in the _lme4_ package in _R_ with VPD as the response variable and temperature as a covariate. We also examined the relationship between temperature and the variation in nightly VPD because physical expectations suggest VPDs should become more variable with warming air temperature. For our random effects, we nested an iButton within the year the recording was taken. The nested random effects accounted for abiotic differences between locations and years due to local topographic effects and interannual seasonality. We generated starting values for the non-linear simulations using the _nlsLM()_ function in the _minpack_ package in _R_. We assessed model performance by evaluating the confidence intervals of the parameter estimates and 95% confidence intervals of the non-linear models.

**Differential gene expression**. We conducted a differential gene expression (DGE) analysis using _DESeq2_ library to identify differentially expressed genes between treatments[75]. We tested for DGE by comparing the gene expression between samples collected at the end of the experiment using the _DESeqDataSetFromMatrix_ function in _DESeq2_. We explored the number of differentially expressed genes for each treatment using two significance thresholds ($\alpha = 0.1$ and $0.001$) on gene-specific $p$-values adjusted for false discovery rate using the default Benjamini–Hochberg procedure[75]. Varying the significance threshold provided insight into broad differences in gene ontology under the most liberal threshold, whereas the more stringent threshold was used to identify candidate genes involved in plasticity.

**Gene ontology enrichment**. We conducted a gene ontology (GO) enrichment analysis on differentially expressed genes using the _GOseq_ package in _R_ (v. 3.4.2) to identify overrepresentation of certain GO terms[76]. The _GOseq_ package determines enrichment of specific GO terms by incorporating the probability a gene will be differentially expressed based upon length of the gene. The _GOseq_ procedure uses a random selection process to evaluate the GO category membership and enrichment by randomly sampling the given dataset to generate a suitable null distribution. Upon identifying the distribution, _GOseq_ evaluates each GO term for under- and overrepresentation. We conducted GO term enrichment on the DGE analysis and the weighted gene co-expression analysis. For enrichment analyses, we removed GO terms with less than 10 and more than 500 genes within a category to reduce enrichment sensitivity and redundancy[77].

**Weighted gene co-expression analysis**. We identified suites of correlated gene networks using weighted gene co-expression network analysis (WGCNA) to improve our understanding of the gene networks and genes underlying acclimation of $r_i$. These analyses were conducted independently of the DGE analysis. We used WGCNA to identify networks of genes (or modules) associated with plasticity of $r_i$. We refer to this analysis as the *skin resistance analysis*. We also conducted an additional analysis to identify potential mechanisms underlying the relationship between mass and temperature within modules identified in the skin resistance analysis, referred to as the *interaction analysis*. We designed these analyses to identify the possible genetic changes underlying specific physiological responses from the experiments.

**Skin resistance analysis**. We began the *skin resistance analysis* using all of the samples from the experimental treatments ($n = 32$). We used the post trimmed and cleaned samples generated from RSEM (from above) to begin the analysis. We then removed any genes that had <10 counts across 28 samples[37], which resulted in a total of 9346 genes for downstream analyses. Gene counts were then normalized using the *varianceStabilizingTransformation* function in DESeq2[75]. We ensured that the dataset did not have any missing data or zero-variance genes, and we removed one sample identified as an outlier based upon cluster analysis prior to constructing co-expression networks[37].

Gene co-expression networks were constructed by building pairwise Pearson correlations between each pair of genes using the *blockwiseModules* function in WGCNA. The resulting co-expression similarity values were transformed into an adjacency matrix using the soft thresholding approach, which favors stronger correlations over weaker correlations between genes. We selected a soft threshold of 4 to simultaneously maximize the model fit of the scale-free topology ($R^2 > 0.80$) and mean connectivity among genes (Supplementary Fig. 2). Then, we defined a dendrogram of the gene networks using average linkage hierarchical clustering coupled with a gene dissimilarity matrix. Finally, we used the Dynamic Tree Cut approach to merge highly correlated modules using a height-cut of 0.25 (ref. [37]). These modules could then be used to understand important networks and genes underlying acclimation of $r_i$.

Module construction can be leveraged to identify biologically meaningful genes or groups of genes underlying variation in a phenotype. We summarized expression for each module using a principle component analysis (PCA) to calculate eigengenes using the *blockwiseModules* function, which summarized the expression for each sample using the first principle component (PC1) for each module. We then used a Pearson correlation to test for associations between eigengene values and the change in $r_i$ for each module using the *cor* function in WGCNA. The resulting *p*-values were estimated using the *corPalueStudent* function in WGCNA. Modules significantly associated with the change in $r_i$, referred to as *skin resistance modules*, also identified potential genes based upon intramodular connectivity[37]. Hub genes are defined as having high intramodular connectivity (or module membership (*MM*)), and previous research has revealed important biological functions by assessing hub genes[37]. We explored the function of hub genes in subsequent analyses based upon their connectivity score, statistical significance ($\alpha < 0.05$) with a particular phenotype, and their association with enriched GO terms. These methods helped to discover potential genes underlying plasticity of water loss physiology.

Temperature and humidity might also represent important cues that explain variation within gene co-expression networks. We evaluated the importance of our treatments in affecting gene expression in skin resistance modules using regression analyses. We used the eigengene values from PC1 as our dependent variable, the change in $r_i$ as a covariate, and temperature and humidity as fixed effects. The model also included interactions with the change in $r_i$ and treatment to determine whether interactions played in important role in affecting gene expression within skin resistance modules. We conducted separate analyses for modules that were negatively and positively associated with $r_i$. Then we used the *car* package to conduct a Type-II ANCOVA to determine whether the treatments influenced gene expression within skin resistance modules.

We identified potential genes underlying plasticity of $r_i$ using two methods. First, we evaluated GO term enrichment based upon the genes within modules significantly associated with $r_i$. We reported enrichment for biological processes, cell components, and molecular function for any terms related to the regulation of water loss. These GO terms included lipid metabolism, skin barrier formation, vasoconstriction, angiogenesis, and lipid excretion. We then cross-referenced these GO terms with hub genes significantly associated with $r_i$ (as denoted by gene significance and intramodular connectivity) to identify potential genes underlying acclimation of $r_i$. We also identified hub genes (from the WGCNA analysis) that significantly correlated with the change in $r_i$ that were also associated with GO terms related to water loss.

**Interaction analysis**. We conducted an additional analysis to explore the interaction between mass and temperature on acclimation of $r_i$. For the interaction analysis, we analyzed the interaction between mass and temperature for all genes within the skin resistance modules. We used linear regression analyses with the transformed gene expression value as the dependent variable, mass as a covariate, and temperature as a factor. We also included the interaction between mass and temperature. Then we conducted Type-II ANCOVA using the *car* package to

evaluate significance of the interaction for each gene by looping across each gene in R. Genes from the skin resistance modules that exhibited a significant interaction between temperature and mass are referred to as *interaction genes*. We assessed the interaction genes for their potential role in water loss regulation by evaluating GO terms associated with each known gene, similar to our previous procedure for identifying candidate genes.

**Gene set enrichment analysis**. We performed a Gene Set Enrichment Analysis (GSEA v.2.1.0) to identify predefined gene sets that showed significant, concordant differences in expression between temperature and humidity treatments[78]. GSEA identifies gene *sets* that show functionally related, but potentially smaller, changes in expression. A custom GSEA database, with each set containing at least 15 genes, was created from GO terms and enzyme code annotations of the assembled transcriptome. We identified enriched or depleted gene sets under specific temperature and humidity conditions using a false discovery rate (FDR) of 0.1. We also determined gene overlap between WGNA, DGE, and GSEA analyses using the *merge()* function in R.

**Reporting summary**. Further information on research design is available in the Nature Research Reporting Summary linked to this article.

## Data availability
We provided the data required to interpret, replicate, and build upon our findings. We provided the raw data on the Open Science Framework repository (https://osf.io/rnsmk/). The data are referenced using the following identifier: https://doi.org/10.17605/OSF.IO/RNSMK. Figure 2 requires the available data from http://www.climatologylab.org/products.html. RNA-seq data is currently available for download on Genbank (BioProject: PRJNA509078).

## Code availability
The Open Science Framework repository (https://osf.io/rnsmk/) contains the annotated script and analysis for Fig. 2 written in Python (v. 3.6).

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

## Acknowledgements

We thank Miranda Gray for her support in the sequencing process, Damien Wilburn for his advice, and Molly Womack for her comments on the manuscript. I also thank the Grant-In-Aid program at the Highlands Biological Research Station, Clemson University, and the Biological Sciences Department at Clemson for financial support. We also thank the Biological Sciences Department and Clemson University Libraries for providing funds to publish the manuscript. This research was funded by the Doctoral Dissertation Improvement Grant (#1601485) through the National Science Foundation's Division of Environmental Biology.

## Author contributions

E.A.R. conducted the experiments, extracted the RNA, prepared the cDNA libraries, performed the analyses, and wrote the manuscript. E.Y.R. and C.E.W. constructed the de novo transcriptome. C.E.W. conducted the GSEA analysis. K.R.Z., C.E.W. and M.W.S. helped to design the experiments and helped with the manuscript preparation.

## Additional information

**Competing interests:** The authors declare no competing interests.

