## [Peer Review File · Nature Communications]

Reviewers' Comments:

Reviewer #1:

Remarks to the Author:

This study presents novel insights into the mechanisms by which ectothermic species sense shifts in their abiotic environment and subsequently respond. The results here are novel in that they upturn what has traditionally been regarded as a simple and straightforward assumption about the cues that initiate phenotypic response. Mainly, they find that changes in environmental humidity do not directly elicit adaptive acclimation responses in hydric physiology, but these responses seem to be indirectly cued through temperature. Additionally, using thorough analyses of transcriptomic data, they elucidate potential mechanisms by which this acclimation response occurs. Lastly, the authors provide a global perspective about the implications of their findings by assessing the reliability of temperature as cue for vapor pressure deficit on local and global scales. Together, these data provide novel and exciting results that are timely as we collectively try to understand the relationships between organisms and their abiotic environment in a world of rapid climatic change. That being said, I believe there are several issues that need to be addressed throughout the manuscript, which I have outlined below. If these issues are met, I believe this study would be a valuable contribution to Nature Communications.

Line 48-50: Behavioral and physiological plasticity are often reversible. But morphological changes, particularly developmental changes often are not. Additionally, it is unclear how an individual would undergo genetic changes in this context. Slight rephrasing necessary

Lines 50-52: plasticity can also be energetically demanding. Is there direct evidence that dispersal across environments is more energetically demanding and plastic responses within environments?

Line 65: At least one citation for the animal example should be included

Line 109-111 and 387-484: Are the temperatures and VPDs used in this context ecologically relevant? Why were these specific values chosen?

Line 117-118: It is unclear how the previous sentence translates into this one

Line 125-126: This is a fascinating and highly unexpected result.

Line 130-131: Should include software for de novo assembly as done for filtering further along in this paragraph

Line 143: The header of this section seems slightly misleading, as temperature seems to be the primary driver plasticity, not humidity

Line 146-148: How many genes were differentially expressed in the other treatments?

Line 151-155: P-values should be added to GO IDs to quantify significance of overrepresentation

Lines 143-232: The authors present a series of thoroughly interrogated GO ontology analyses that together, make the link between skin resistance to water loss and potential subordinate physiological mechanisms – and link some of those mechanisms to thermal cues. However, with the given data, the authors have an opportunity to further understand the mechanisms that drive plastic response of regulatory networks to external cues. For instance, how are individual DE genes clustered among the

identified regulatory modules? Does the connectivity of DE genes within their respective modules support plasticity driven by primary through hub genes or through less pleiotropic alterations across peripheral genes? Or neither? In my opinion, these additional analyses would greatly add to the novelty of the study.

Additionally, given how well curated the vertebrate angiogenesis pathway is (and potentially lipid barrier formation?), the authors may consider bolstering their de novo approach to identifying the regulatory architecture of the tissues (via WGCNA) with discrete analyses of the genes within known pathways that are overrepresented in the analyses presented.

Lines 274-278: It is unclear to me whether the variation in plasticity observed here is due to 1) individual variation in plastic ability under extreme challenge (some individuals have high resistance and low plasticity, while others have low initial resistance but high plasticity) or 2) whether the conditions presented resulted in coalescence of individual phenotypes around a physiological optimum. In the latter case, individuals with higher initial skin resistance may have been close to the new optimum initially and therefore display lesser plasticity to move to the optimum. In that case, those individuals could be similarly plastic if the new acclimation environment was sufficiently stressful to shift the phenotypic optimum away from their initial set point. I think this would be clearer if the authors gave more information about the ecological relevance of their acclimation treatments.

Lines 297-298: Are these loss of function mutations?

Lines 321-322: This sentence is unclear.

Lines 330 and 334: "genetic" should be replaced with "regulatory"

General comment: I don't recall the authors making any explicit connections between the genes involved in the plastic response to temperature/humidity and those that have been studied in the context of amphibian limb regeneration. This connection should be directly assessed to bolster Lines 31-33 and 325-326.

Reviewer #2:

Remarks to the Author:

In this manuscript, authors collected physiological changes of salamander against different temperature and humidity, and sought molecular mechanisms of these physiological changes. They obtained huge ecological data, and successfully found differentially expressed genes reflecting physiological changes. The question authors proposed is very challenging and interesting topic of ecology and evolutionary biology. However, unfortunately, several points of the manuscript are overstatement. I think the current version of manuscript is not suitable for publication for Nature Communications and recommend reject or major revision.

Authors states physiological phenotypic plasticity they found is linked to tissue regeneration of vasculature. I think such statement is based on insufficient logic. Although they measured water loss rates of salamanders at different environment and found differences of water loss rate depending on environment, they never described histological and morphological changes linked to such physiological changes. In general, to proof tissue regeneration, histological description of tissue reconstruction is needed. Thus, it is hard to say that such physiological changes are linked to tissue regeneration at current situation.

Differentially expressed genes (DEGs) against changes of temperature and humidity are identified in this manuscript. Based on the list of these DEGs, authors discuss the molecular mechanism of physiological phenotypic plasticity. However, they just identified correlations between gene expressions and environmental changes and never validated functional contribution of these genes for expression of phenotypic plasticity. Because their discussions are based on such descriptive results, some points of them are overstatement and doubtful. I think when studies about phenotypic plasticity are published for high impact journal such as nature communications, it is important to show detailed molecular mechanisms based on molecular experiments. If such solid evidences are obtained, people can discuss ecological influence of specific phenotypic plasticity effectively.

Minor comments:

Lines 98-99, 114

What is "star method"?

Line 146

Fig. 2 => Fig. 3?

Lines 181-182

Authors should show gene lists.

Lines 265-266

This expression is very strange. Authors studied neither embryonic development nor angiogenesis in this manuscript.

Lines 270-272

These contexts were also strange logic.

Line 361

Experiment procedures are very complicated. I recommend adding figure to explain scheme of experiments.

Line 487

"Adequate sequencing depth" is ambiguous. Authors should show specific read number of each sample.

Lines 497-498

This is confusing. Although authors only described the sampling of salamander skin tissues, they used skin and heart samples for de novo transcriptome assembly. How did they obtain heart samples???

Lines 561-562

Why did they remove these samples?

Line 592

I could not find Figure S1.

Reviewer #3:

Remarks to the Author:

A novel study that aims to identify the molecular mechanisms underlying plasticity in skin resistance to desiccation. This study brilliantly integrates modern molecular techniques with physiological experiments and ecological thinking. I truly admire the careful experimental approach used to achieve ambitious goals. The authors performed an acclimation experiment to tease apart the roles of temperature and humidity as environmental cues for plasticity of skin resistance to water loss. By extracting RNA from salamanders before and after the experiment, they were able to quantify differential gene expression throughout the experiment among experimental treatments (warm/dry, warm/wet, cool/dry, and cool/wet). The authors include an additional analysis of global climate data to suggest that temperature is a reliable cue for desiccation risk across large spatial scales, thus providing broad applications of their findings. This is an area of biology that is poorly understood, and this paper contributes to our understanding of species' responses to climate change. Overall, the manuscript is well-written and will have a high impact on a broad audience.

However, I have two major conceptual issues with the interpretation of the results and one methodological issue.

Conceptual concerns:

First, I am not convinced that temperature is a strong enough predictor of VPD at the representative field sites to act as a reliable stimulus to prevent, or reduce, desiccation. On line 96, the authors state that "Salamanders experienced predictable variation in the near-surface temperatures and VPDs (Fig. 1A)". Yet, the data seem to show that temperature is a very weak predictor of both the standard deviation of ($R^2 = 0.05$) and mean VPD ($R^2 = 0.09$). The significant p-value in these cases are likely due to the immense sample sizes used for the analysis, rather than being a biologically meaningful predictor. Further, even if temperature are a strong predictor of VPD, plethodontid salamanders are known to limit behavioral activity in warm and dry environments, thus reducing their exposure to the selective pressures that would be required to drive the evolution of r_i plasticity suggested by the authors. My concern is that the authors base the majority of their analyses on temperature being a reliable cue of VPD, since VPD did not have a significant effect on r_i in the acclimation experiment, but provide evidence (Fig 1) that suggests otherwise.

Second, I am concerned that the authors have not considered likely alternatives to the mechanisms driving the differential gene expression they observed throughout the acclimation experiment. The authors conclude that their findings are "highly suggestive of a role of blood vessel development in regulating resistance to desiccation" (line 261-262). I am concerned that the major changes in gene expression were found only in the warm and wet treatment but were absent in the warm and dry treatment. If the upregulation of these genes was in fact tied to functional changes that reduce r_i , why wouldn't salamanders in the warm, dry treatment exhibit similar physiological responses? It is this question, that leads me to believe that the observed functional response is likely linked with shedding rate and/or oxygen regulation rather than desiccation.

Salamanders shed more regularly in warmer temperatures. I'm unaware of studies that examine the relationship between shed rate and moisture in amphibians, but reptiles are known to soak in water prior to a bout of ecdysis. Thus, I think it is likely that the salamanders would have experienced a higher shed rate in the warm and wet treatment than the warm and dry treatment. If true, an increase in shedding and skin regeneration may explain the observed change in gene expression that is tied to skin integrity throughout the experiment.

Similarly, a change in the rate of oxygen diffusion across epithelial membranes may have driven the observed physiological responses. I was very surprised that the authors did not mention oxygen uptake, given that this species is lungless and relies on cutaneous respiration. Salamanders in the warm treatments would have a greater demand for oxygen to keep up with an increase in metabolic

expenditure. Since they reduced the permeability of their skin, the diffusion rate of oxygen is likely to be slower. Thus, it seems likely that salamanders would respond by increasing vascularization to increase the surface area available for oxygen diffusion and smooth muscle cell proliferation to increase the flow of blood throughout the tissues. In this case, I'm not sure why the physiological response would only occur in warm and wet, rather than warm and dry environments. I understand that the skin resistance and interaction analyses were meant to convince the reader that the observed gene expression was related to plasticity in r_i , but very little variation in plasticity was actually explained by the modules. Thus, I don't think the authors can rule out the alternative explanations.

Methodological concern:

Given that body mass had a major effect on r_i , it seems like the authors should have accounted for mass in their calculation of r_i . Surface area:volume ratio is critical in determining diffusion rates and osmotic exchange across the skin and should have been included in the analysis rather than treating mass as a covariate. I believe that including mass in the calculation of r_i (so that the units are $s/cm/g$) would drastically change the interpretation of the experimental results which is the centerpiece of this paper. After doing a rough simulation, it looks as though the interactive effect of temperature and mass on change in r_i might disappear. Change in skin resistance would decrease with size in both temperature treatments rather than increasing with size in the warm treatment as is shown in Fig. 2b. I believe that the corrected relationship makes more biological sense because small individuals have a larger surface area:volume ratio and are at a higher risk of desiccation in warm temperatures. Thus, a greater change in r_i should provide greater benefit to smaller individuals, and might be under stronger selection.

Aside from these issues, which I believe can be addressed by restructuring the narrative and modifying the analyses, I believe this paper will be highly impactful to a wide audience.

Reviewers' comments:

Reviewer #1 (Remarks to the Author):

This study presents novel insights into the mechanisms by which ectothermic species sense shifts in their abiotic environment and subsequently respond. The results here are novel in that they upturn what has traditionally been regarded as a simple and straightforward assumption about the cues that initiate phenotypic response. Mainly, they find that changes in environmental humidity do not directly elicit adaptive acclimation responses in hydric physiology, but these responses seem to be indirectly cued through temperature. Additionally, using thorough analyses of transcriptomic data, they elucidate potential mechanisms by which this acclimation response occurs. Lastly, the authors provide a global perspective about the implications of their findings by assessing the reliability of temperature as cue for vapor pressure deficit on local and global scales. Together, these data provide novel and exciting results that are timely as we collectively try to understand the relationships between organisms and their abiotic environment in a world of rapid climatic change. That being said, I believe there are several issues that need to be addressed throughout the manuscript, which I have outlined below. If these issues are met, I believe this study would be a valuable contribution to Nature Communications.

Line 48-50: Behavioral and physiological plasticity are often reversible. But morphological changes, particularly developmental changes often are not. Additionally, it is unclear how an individual would undergo genetic changes in this context. Slight rephrasing necessary

Response: We rephrased the sentence by removing the “rapid and reversible” phrase, and we were more specific as to what we meant in the following sentence (line 52).

Lines 50-52: plasticity can also be energetically demanding. Is there direct evidence that dispersal across environments is more energetically demanding and plastic responses within environments?

Response: The reviewer brings up an interesting idea. Several theoretical analyses by Schoener *et al.* have explored the interaction between dispersal and costs of plasticity, but to our knowledge, we have not seen an explicit comparison of the costs between plasticity and dispersal. We believe that few have tackled this problem because of the diverse types of costs imposed by either dispersal or plasticity, which could include energetic expenditure, lost opportunity costs, or greater exposure to predation. Our point here, however, was to illustrate that plasticity allows organisms to avoid the costs of dispersal altogether, absent of the potential costs that might occur from plasticity. That being said, we switched the citation to a review that outlines the diverse array of costs associated with dispersal. We believe readers can refer to this citation for a greater understanding of the diverse costs of dispersal (line 55).

Line 65: At least one citation for the animal example should be included

Response: Animals are included after the “and animals under climate change” (line 70).

Line 109-111 and 387-484: Are the temperatures and VPDs used in this context ecologically relevant? Why were these specific values chosen?

Response: Yes, these conditions were selected based upon the conditions that salamander experience under contemporary conditions (cool cycle) and under conditions they might experience under climate warming (the warm cycle). We added this detail in the methods describing the acclimation experiment (line 450).

Line 117-118: It is unclear how the previous sentence translates into this one

Response: We agree. We adjusted the phrasing of the sentences to make this clearer.

Line 125-126: This is a fascinating and highly unexpected result.

Response: Thank you.

Line 130-131: Should include software for de novo assembly as done for filtering further along in this paragraph

Response: Included – thank you.

Line 143: The header of this section seems slightly misleading, as temperature seems to be the primary driver plasticity, not humidity

Response: We agree. However, we changed this analysis to understand differential gene expression due to temperature and VPD in isolation, not their interaction as was previously reported. We decided to change the analysis because the interaction between temperature and VPD was never important in our phenotype of interest, resistance to water loss (line 370). Salamanders may use humidity for other purposes, but commenting on these purposes is beyond the scope of this manuscript. Moreover, the DGE analysis is now far more consistent (and thus comparable) to the phenotypic analysis and the additional gene expression analysis. The title is also now been updated (line 164).

Line 146-148: How many genes were differentially expressed in the other treatments?

Response: We added these details (lines 165 -170) and the information is available in Tables 2, 3 and 4 of the supplement.

Line 151-155: P-values should be added to GO IDs to quantify significance of overrepresentation

Response: We added the statistics, but there were several GO terms that were mentioned citing specific genes of interest that were not necessarily enriched (though their function was very similar to enriched GO terms). In these cases, it was not possible to add a p-value.

Lines 143-232: The authors present a series of thoroughly interrogated GO ontology analyses

that together, make the link between skin resistance to water loss and potential subordinate physiological mechanisms – and link some of those mechanisms to thermal cues. However, with the given data, the authors have an opportunity to further understand the mechanisms that drive plastic response of regulatory networks to external cues. For instance, how are individual DE genes clustered among the identified regulatory modules? Does the connectivity of DE genes within their respective modules support plasticity driven by primary through hub genes or through less pleiotropic alterations across peripheral genes? Or neither? In my opinion, these additional analyses would greatly add to the novelty of the study.

Response: We agree. We compared overlap in the genes among the three different gene expression analyses (DGE, WGCNA, and GSEA). Our overlap analysis, inspired by your comment, revealed consistent patterns between expression analyses and identified some additional potential candidate genes. These genes are listed in Table S12, TableS13, and Table S14. Overall, comparing the overlap between GSEA and the other two analyses revealed a high degree of consistency in the mechanisms proposed in our study. The comparison also revealed important hub genes from the WGCNA analysis that were found in the GSEA. We thank the reviewer very much for this request as we believe it substantially improved the manuscript.

Additionally, given how well curated the vertebrate angiogenesis pathway is (and potentially lipid barrier formation?), the authors may consider bolstering their de novo approach to identifying the regulatory architecture of the tissues (via WGCNA) with discrete analyses of the genes within known pathways that are overrepresented in the analyses presented.

Response: We conducted an additional analysis to address this question specifically, called gene set enrichment analysis or GSEA. For the description of GSEA, please see our methods on line 737. The GSEA analysis provided substantial support for our interpretations by identifying suites of known pathways corresponding to regulation of angiogenesis in response to warm temperatures. More specifically, we were able to bolster support and improve our explanation for the mechanisms underlying plasticity in r_i .

Lines 274-278: It is unclear to me whether the variation in plasticity observed here is due to 1) individual variation in plastic ability under extreme challenge (some individuals have high resistance and low plasticity, while others have low initial resistance but high plasticity) or 2) whether the conditions presented resulted in coalescence of individual phenotypes around a physiological optimum. In the latter case, individuals with higher initial skin resistance may have been closure to the ne optimum initially and therefore display lesser plasticity to move to the optimum. In that case, those individuals could be similarly plastic if the new acclimation environment was sufficiently stressful to shift the phenotypic optimum away from their initial set point. I think this would be clearer if the authors gave more information about the ecological relevance of their acclimation treatments.

Response: We hope that the newly included comments on the ecological relevance of the experimental conditions clarifies this issue. We find option 2 to be unlikely because the temperatures and humidities are relevant to conditions salamanders experience under stressful conditions (i.e., the extreme climate warming scenario). We also think that the low plastic, high resistance phenotype was very surprising given their exposure to moist conditions for a month

prior to the experiment. Because they maintained this phenotype throughout the acclimation period and highly exposure to hot, dry conditions, we should have detected a plastic response if it existed. Exposure to conditions beyond this range would have likely been lethal. The wide range of conditions experienced by the salamanders combined with their continued low plasticity phenotype is highly suggestive that these individuals lack the ability to adjust water loss rates.

Lines 297-298: Are these loss of function mutations?

Response: Yes, we added this clarification.

Lines 321-322: This sentence is unclear.

Response: We removed this sentence.

Lines 330 and 334: “genetic” should be replaced with “regulatory”

Response: Done.

General comment: I don't recall the authors making any explicit connections between the genes involved in the plastic response to temperature/humidity and those that have been studied in the context of amphibian limb regeneration. This connection should be directly assessed to bolster Lines 31-33 and 325-326.

Response: With the new GSEA analysis, we made direct connections to pathways involved in the regulation of *de novo* blood vessel formation. We found evidence for conserved pathways involved in development processes and angiogenesis with the new GSEA analysis. These greatly improve our ability to identify the mechanisms underlying the plasticity in resistance to water loss.

Reviewer #2 (Remarks to the Author):

In this manuscript, authors collected physiological changes of salamander against different temperature and humidity, and sought molecular mechanisms of these physiological changes. They obtained huge ecological data, and successfully found differentially expressed genes reflecting physiological changes. The question authors proposed is very challenging and interesting topic of ecology and evolutionary biology. However, unfortunately, several points of the manuscript are overstatement. I think the current version of manuscript is not suitable for publication for Nature Communications and recommend reject or major revision.

Authors states physiological phenotypic plasticity they found is linked to tissue regeneration of vasculature. I think such statement is based on insufficient logic. Although they measured water loss rates of salamanders at different environment and found differences of water loss rate depending on environment, they never described histological and morphological changes linked to such physiological changes. In general, to proof tissue regeneration, histological description of tissue reconstruction is needed. Thus, it is hard to say that such physiological changes are linked to tissue regeneration at current situation.

Differentially expressed genes (DEGs) against changes of temperature and humidity are identified in this manuscript. Based on the list of these DEGs, authors discuss the molecular mechanism of physiological phenotypic plasticity. However, they just identified correlations between gene expressions and environmental changes and never validated functional contribution of these genes for expression of phenotypic plasticity. Because their discussions are based on such descriptive results, some points of them are overstatement and doubtful. I think when studies about phenotypic plasticity are published for high impact journal such as nature communications, it is important to show detailed molecular mechanisms based on molecular experiments. If such solid evidences are obtained, people can discuss ecological influence of specific phenotypic plasticity effectively.

Response: We agree that further experiments at the molecular and cellular level are required to validate that vasculature regeneration is involved. We provided more clarity that these results represent intriguing hypotheses on the role of tissue regeneration in physiological plasticity in the first two paragraphs of the discussion. We also explicitly mention that determining the exact mechanisms driving physiological plasticity would require further experiments at the cellular and molecular level (line 336).

Minor comments:

Lines 98-99, 114

What is “star method”?

Response: A typo. Corrected. Thank you.

Line 146

Fig. 2 => Fig. 3?

Response: Corrected. Thank you.

Lines 181-182

Authors should show gene lists.

Response: We added the new table (now Table S5).

Lines 265-266

This expression is very strange. Authors studied neither embryonic development nor angiogenesis in this manuscript.

Response: We understand the confusion. We rephrased our interpretations of the data extensively in hopes of making these connections clearer. Please see the first paragraph of the discussion.

Lines 270-272

These contexts were also strange logic.

Response: We hope that the revisions improved the logic of our manuscript by making better connections between the mechanisms (the potential targets of selection) and the greater selection pressure caused by warming and drying under future climate change.

Line 361

Experiment procedures are very complicated. I recommend adding figure to explain scheme of experiments.

Response: We added a figure in supplement (now Figure S1 based on the new order of the supplemental material.)

Line 487

“Adequate sequencing depth” is ambiguous. Authors should show specific read number of each sample.

Response: We agree. The sentence was supposed to address the trade-off between sequencing depth and sample size. We revised the sentence to be more clear (line 574).

Lines 497-498

This is confusing. Although authors only described the sampling of salamander skin tissues, they used skin and heart samples for de novo transcriptome assembly. How did they obtain heart samples???

Response: We added sentences in the methods describing how these tissues were sampled and why we did not include analyses on these samples (line 551). Essentially, we collected heart samples for other purposes beyond the scope of this manuscript. As you mentioned, the methods are already highly complex and adding these results would confuse the narrative.

Lines 561-562

Why did they remove these samples?

Response: These steps are performed to reduce sensitivity to GO terms with only a few associated genes (which increases the chance of being under- or overexpressed) and redundancy of genes across multiple GO terms. The procedure is a commonly used default option in GO term enrichment analyses. We provided a citation that reviews these issues (line 658).

Line 592

I could not find Figure S1.

Response: Corrected. Thank you.

Reviewer #3 (Remarks to the Author):

A novel study that aims to identify the molecular mechanisms underlying plasticity in skin resistance to desiccation. This study brilliantly integrates modern molecular techniques with

physiological experiments and ecological thinking. I truly admire the careful experimental approach used to achieve ambitious goals. The authors performed an acclimation experiment to tease apart the roles of temperature and humidity as environmental cues for plasticity of skin resistance to water loss. By extracting RNA from salamanders before and after the experiment, they were able to quantify differential gene expression throughout the experiment among experimental treatments (warm/dry, warm/wet, cool/dry, and cool/wet). The authors include an additional analysis of global climate data to suggest that temperature is a reliable cue for desiccation risk across large spatial scales, thus providing broad applications of their findings. This is an area of biology that is poorly understood, and this paper contributes to our understanding of species' responses to climate change. Overall, the manuscript is well-written and will have a high impact on a broad audience.

However, I have two major conceptual issues with the interpretation of the results and one methodological issue.

Conceptual concerns:

First, I am not convinced that temperature is a strong enough predictor of VPD at the representative field sites to act as a reliable stimulus to prevent, or reduce, desiccation. On line 96, the authors state that "Salamanders experienced predictable variation in the near-surface temperatures and VPDs (Fig. 1A)". Yet, the data seem to show that temperature is a very weak predictor of both the standard deviation of ($R^2 = 0.05$) and mean VPD ($R^2 = 0.09$). The significant p -value in these cases are likely due to the immense sample sizes used for the analysis, rather than being a biologically meaningful predictor. Further, even if temperature are a strong predictor of VPD, plethodontid salamanders are known to limit behavioral activity in warm and dry environments, thus reducing their exposure to the selective pressures that would be required to drive the evolution of r_i plasticity suggested by the authors. My concern is that the authors base the majority of their analyses on temperature being a reliable cue of VPD, since VPD did not have a significant effect on r_i in the acclimation experiment, but provide evidence (Fig 1) that suggests otherwise.

Response: We agree that our analyses in the previous version did not present a strong case that temperature represents a "good predictor" of VPD. We changed our analyses to control for variation associated with local site-level effects and inter-annual seasonality. We opted to use non-linear, mixed effects models to account for the aforementioned variables and capture the non-linearity in the dataset. We used standard errors of the parameter estimates and 95% confidence intervals of the model to evaluate model performance because R^2 values are not appropriate for non-linear models and p -values can be misleading, as noted by this reviewer.

We also focused specifically on conditions that our species of salamander prefers based upon experiments by Spotila (1972). Finally, we also analyzed the raw VPD data, instead of nightly averages. Averaging across a night produced erroneous results by hiding important variation in VPDs at high temperatures. The new analysis provides a much richer and complex investigation of the relationship between temperature and VPD at the local scale. The new analysis identified an exponential relationship between temperature and VPD – a pattern that is highly consistent with theory (Stull 2010) and global empirical trends presented in our manuscript.

We also want to clarify our thought process with the reviewer. We based our analyses on temperature being a reliable predictor of VPD because of the inherent physical effect of temperature on saturation vapor pressures (Stull 2010) and the variability of VPD in nature. We included the relationship between temperature and saturation vapor pressure to illustrate why we would expect VPD to increase non-linearly with temperature (Figure 1a). In nature, VPDs are too variable to use it as a cue for plasticity. Temperature, however, may be a reliable predictor of the potential for desiccation risk due to its correlation to high and more variable VPDs, especially when temperatures begin to exceed 20°C (new exponential analysis). Moreover, the physiological plasticity in response to temperature but not VPD supports our hypothesis (this study) and in previous experiments (Riddell et al. 2015). Based on the physical relationship between temperature and VPD, variability in the field, and physiological responses in the lab, we believe we have provided extensive support for the utilization of temperature as a cue to predict desiccation risk.

Second, I am concerned that the authors have not considered likely alternatives to the mechanisms driving the differential gene expression they observed throughout the acclimation experiment. The authors conclude that their findings are “highly suggestive of a role of blood vessel development in regulating resistance to desiccation” (line 261-262). I am concerned that the major changes in gene expression were found only in the warm and wet treatment but were absent in the warm and dry treatment. If the upregulation of these genes was in fact tied to functional changes that reduce r_i , why wouldn't salamanders in the warm, dry treatment exhibit similar physiological responses? It is this question, that leads me to believe that the observed functional response is likely linked with shedding rate and/or oxygen regulation rather than desiccation.

Salamanders shed more regularly in warmer temperatures. I'm unaware of studies that examine the relationship between shed rate and moisture in amphibians, but reptiles are known to soak in water prior to a bout of ecdysis. Thus, I think it is likely that the salamanders would have experienced a higher shed rate in the warm and wet treatment than the warm and dry treatment. If true, an increase in shedding and skin regeneration may explain the observed change in gene expression that is tied to skin integrity throughout the experiment.

Response: We agree that shedding rate is an important consideration. However, recent studies on anurans uncovered that water loss rates are positively associated with shedding rates (Russo et al. 2018). The increase in water loss rates would translate into lower resistances to water loss simply based on the equation for resistance to water loss:

$$\text{Resistance to water loss} = \text{water vapor density gradient/cutaneous water loss rate}$$

We understand that previous studies have found shedding rate to increase in warm temperatures, but this would have coincided with a reduction in skin resistance to water loss. Because the increase in skin resistance coincided with the down-regulation of regeneration pathways, we find it highly unlikely that shedding confounds our results. Based on this comment and the next, we also believed that we were not clear enough on the hypothesized mechanism involving regulation of water loss. We added a detailed description of the hypothesized mechanisms regulating resistance to water loss in the first paragraph of the discussion. We hope that this provides

greater clarity to our manuscript.

Similarly, a change in the rate of oxygen diffusion across epithelial membranes may have driven the observed physiological responses. I was very surprised that the authors did not mention oxygen uptake, given that this species is lungless and relies on cutaneous respiration. Salamanders in the warm treatments would have a greater demand for oxygen to keep up with an increase in metabolic expenditure. Since they reduced the permeability of their skin, the diffusion rate of oxygen is likely to be slower. Thus, it seems likely that salamanders would respond by increasing vascularization to increase the surface area available for oxygen diffusion and smooth muscle cell proliferation to increase the flow of blood throughout the tissues. In this case, I'm not sure why the physiological response would only occur in warm and wet, rather than warm and dry environments. I understand that the skin resistance and interaction analyses were meant to convince the reader that the observed gene expression was related to plasticity in r_i , but very little variation in plasticity was actually explained by the modules. Thus, I don't think the authors can rule out the alternative explanations.

Response: The reviewer suggested that salamanders increase vascularization to increase gas exchange to counter the increase in skin resistance. We agree that we cannot completely rule out that the regeneration of vasculature may be related to respiratory regulation. However, an increase in blood flow to the skin, which would be a result of increased vascularization, has been known to increase water loss rates in amphibians (Burggren et al. 2005). Consistent with these results, we found that the reduction in genes associated with angio- and vasculogenesis were negatively associated with skin resistance (meaning, lower expression of these genes were associated with lower water loss rates). Therefore, we find it unlikely that salamanders are compensating for greater skin resistance to water loss by increasing vascularization. We explored this interaction in a previous manuscript (Riddell et al. 2018), and we found that plasticity in water loss rates and metabolic rates were highly coupled — meaning we have limited evidence that they can compensate for the increase in skin resistance.

That being said, we included this explanation in the second paragraph of the discussion. Regardless, regeneration appears to play a role in physiological plasticity. A more likely scenario is that the reduction in blood vessel formation to reduce water loss rates is also inhibiting their ability to breathe across their skin (also mentioned in the discussion). Thus, we believe the mechanisms highlight the link between the two physiological traits and a cost associated with plasticity.

Methodological concern:

Given that body mass had a major effect on r_i , it seems like the authors should have accounted for mass in their calculation of r_i . Surface area:volume ratio is critical in determining diffusion rates and osmotic exchange across the skin and should have been included in the analysis rather than treating mass as a covariate. I believe that including mass in the calculation of r_i (so that the units are $s/cm/g$) would drastically change the interpretation of the experimental results which is the centerpiece of this paper. After doing a rough simulation, it looks as though the interactive effect of temperature and mass on change in r_i might disappear. Change in skin resistance would decrease with size in both temperature treatments rather than increasing with size in the warm treatment as is shown in Fig. 2b. I believe that the corrected relationship makes more biological sense because small individuals have a larger surface area:volume ratio and

are at a higher risk of desiccation in warm temperatures. Thus, a greater change in r_i should provide greater benefit to smaller individuals, and might be under stronger selection.

Response: Mass is already included in the calculations for resistance to water loss. Resistance to water loss is calculated using the cutaneous water loss rate, which has units of grams of water per unit time per unit surface area. In our studies, we estimate surface area of the salamander using established allometric equations for plethodontids. Because of the existing inclusion of mass, dividing resistance by mass would be inappropriate and inconsistent with the literature. That being said, including mass in our statistical model is helpful for understanding variation with mass.

Dividing a physiological rate by mass does not control or correct for mass. Many mass-specific or surface area-specific physiological rates are expected to vary with mass *a priori*. Smaller animals, for instance, should have higher mass- or surface area-corrected water loss rates than larger animals. Thus, even if we divided resistance to water loss by mass, we would still need to account for the mass-specific variation expected based upon first principles (meaning, mass would still need to be in the model). For best practices, we advocate for using mass in the statistical analysis to account for its known impact on physiological rates.

Aside from these issues, which I believe can be addressed by restructuring the narrative and modifying the analyses, I believe this paper will be highly impactful to a wide audience.

Response: Thank you.

Reviewers' Comments:

Reviewer #1:

Remarks to the Author:

My concerns regarding the manuscript have been resolved in its latest iteration. I believe this will be an exciting addition to Nature Communications.

Reviewer #3:

Remarks to the Author:

Thank you for your thoughtful and thorough responses to the initial review. You have resolved my major concerns and I look forward to seeing more studies that follow-up on this work.

REVIEWERS' COMMENTS:

Reviewer #1 (Remarks to the Author):

My concerns regarding the manuscript have been resolved in its latest iteration. I believe this will be an exciting addition to Nature Communications.

Reviewer #3 (Remarks to the Author):

Thank you for your thoughtful and thorough responses to the initial review. You have resolved my major concerns and I look forward to seeing more studies that follow-up on this work.